# Effect of Nano-Y_2_O_3_ Content on Microstructure and Mechanical Properties of Fe18Cr Films Fabricated by RF Magnetron Sputtering

**DOI:** 10.3390/nano11071754

**Published:** 2021-07-05

**Authors:** Bang-Lei Zhao, Le Wang, Li-Feng Zhang, Jian-Gang Ke, Zhuo-Ming Xie, Jun-Feng Yang, Xian-Ping Wang, Ting Hao, Chang-Song Liu, Xue-Bang Wu

**Affiliations:** 1Key Laboratory of Materials Physics, Institute of Solid State Physics, HFIPS, Chinese Academy of Sciences, Hefei 230031, China; zbl1234@mail.ustc.edu.cn (B.-L.Z.); zlf05358@mail.ustc.edu.cn (L.-F.Z.); jingkejk@126.com (J.-G.K.); zmxie@issp.ac.cn (Z.-M.X.); xpwang@issp.ac.cn (X.-P.W.); xbwu@issp.ac.cn (X.-B.W.); 2University of Science and Technology of China, Hefei 230026, China; 3Beijing Institute of Technology Chongqing Innovation Center, Chongqing 401135, China; wangle@bit.edu.cn; 4School of Materials Science and Engineering, Beijing Institute of Technology, Beijing 100081, China; 5Lu’an Branch, Anhui Institute of Innovation for Industrial Technology, Lu’an 237100, China; 6School of Mechanical Engineering, Suzhou University of Science and Technology, Suzhou 215009, China; hao.ting@mail.usts.edu.cn

**Keywords:** FeCr-based films, oxide dispersion strengthening, nanoindentation hardness

## Abstract

In this work, FeCr-based films with different Y_2_O_3_ contents were fabricated using radio frequency (RF) magnetron sputtering. The effects of Y_2_O_3_ content on their microstructure and mechanical properties were investigated through scanning electron microscopy (SEM), transmission electron microscopy (TEM), X-ray photoelectron spectroscopy (XPS), inductive coupled plasma emission spectrometer (ICP) and a nanoindenter. It was found that the Y_2_O_3_-doped FeCr films exhibited a nanocomposite structure with nanosized Y_2_O_3_ particles uniformly distributed into a FeCr matrix. With the increase of Y_2_O_3_ content from 0 to 1.97 wt.%, the average grain size of the FeCr films decreased from 12.65 nm to 7.34 nm, demonstrating a grain refining effect of Y_2_O_3_. Furthermore, the hardness of the Y_2_O_3_-doped FeCr films showed an increasing trend with Y_2_O_3_ concentration, owing to the synergetic effect of dispersion strengthening and grain refinement strengthening. This work provides a beneficial guidance on the development and research of composite materials of nanocrystalline metal with a rare earth oxide dispersion phase.

## 1. Introduction

The discovery and application of nuclear energy are regarded as two of the most important scientific and technological achievements of the twentieth century. As a kind of clean and sustainable energy, nuclear energy can extensively replace conventional fossil energy [1,2]. So far, the studies on nuclear energy systems have made great progress, but material issue is still one of the key challenges that restrict the development of nuclear energy [3,4,5,6]. In particular, for future advanced nuclear reactors, such as fast reactors [7], thermonuclear fusion reactors (TFR) [8] and accelerator driven sub-critical systems (ADS) [9], due to their severely harsh service environment which includes a high temperature, strong irradiation and strong corrosion, materials with superior service performance and reliability are urgently needed.

FeCr-based steels have been widely considered as one of the most important structure materials for nuclear reactors because of their outstanding mechanical properties. However, in the long-term nuclear service process, FeCr-based structural alloys, such as 304 and 316 stainless steel, have to bear high-dose irradiation damages, which accordingly induce irradiation hardening [5], porosity evolution and swelling [10], high temperature helium embitterment (HTHE) [11], etc. Therefore, how to increase the irradiation resistance of FeCr-based alloys is still a key problem to be solved until now. Oxide dispersion-strengthened (ODS) steels are a great development in nuclear energy structure materials, owing to their excellent irradiation resistance property [12,13]. S. Ukai et al. found that nanoparticles could act as sinks for the trapping of helium atoms and point defects, thus retarding radiation-induced material degradation [14]. Moreover, FeCr alloys with the addition of oxide particles have been shown to have high tensile, creep and fatigue strength, have thermal stability through the promotion of radiation-induced defect recombination, trap He atoms and impede dislocation climb and glide [15]. In particular, the addition of a small amount of Y_2_O_3_ nanoparticles (0.1~0.5 wt.%) in the alloy matrix can not only improve its mechanical properties, but also annihilate the defects effectively caused by irradiation [16,17,18,19]. Moreover, the grain refinement of structural materials is also treated as one of the effective methods of optimizing the strength and irradiation resistance properties, owing to the high density of interface, especially in nanocrystalline materials, as reported in nanocrystalline Au (grain size~23 nm) [20] and TiNi alloys (grain size 23–31 nm) [21]. This is because the high-density interface in nanocrystalline materials can effectively absorb and annihilate the irradiation defects, which shows a better anti-irradiation performance than that of coarse grain materials. Meanwhile, an increase in the strength of nanostructured materials has been achieved over the last several decades based on extrapolations of the grain size dependence of mechanical properties of conventional materials [22]. The general enhancement in the materials’ strength is dependent on the grain size, which is often described by the empirical Hall–Petch relationship [23]. 

Magnetron sputtering is known as one of the most useful methods of fabricating nanocrystalline materials [24,25,26,27], in which grain size, density and film thickness can be well controlled through adjusting the sputtering parameters, such as substrate temperature, sputtering power, sputtering time and sputtering pressure. It is known that thin films and bulks materials have different application fields. The thin films cannot be used as the construction materials of nuclear reactors. However, the FeCr-based thin films can be adopted as a platform to model the nanocrystalline metal materials with an extremely high volume fraction of grain boundaries, and to investigate the mechanism of Y_2_O_3_-induced strengthening. Therefore, in this work, the magnetron sputtering ODS-FeCr films were prepared and researched in order to provide a beneficial guidance for the research and development of fission reactor structural materials in the future.

## 2. Experimental

Composite targets composed of Fe_18_Cr-based steel holder and Y_2_O_3_ slices were designed (Figure 1a). By changing the number of Y_2_O_3_ slices, the Y_2_O_3_ contents could be adjusted in the deposited film. Si single crystal wafers with (111) orientation were used as substrates. Y_2_O_3_-doped FeCr films were fabricated by radio frequency (RF) magnetron sputtering technique. The rotating target was used in this deposition experiment to guarantee the homogeneous distribution of chemical elements in the films. There was an annular magnetic field around the target and the electrons moved in a cycloid-like manner on the surface of the target. Before sputtering, the vacuum chamber was evacuated to 8.0 × 10^−4^ Pa by a molecular pump, and then the working argon (Ar) gas was filled into the chamber and the working pressure was kept at about 1.0 Pa. The sputtering time was 4 h, the sputtering power was controlled at 70 W and the substrate temperature was fixed at 200 °C.

The crystalline structure of the films was characterized using a grazing incident X-ray diffractometer (GIXRD, PANalytical Company, Netherlands, Philips X’pert PRO, Cu Kα radiation, wavelength ~0.15418 nm) with a step size of 0.06° and grazing angle of 1.5°. The surface and cross-section morphology of the films were observed by field emission scanning electron microscopy (FESEM, Sirion 200FEG, HITACHI, Japan, accelerating voltage 5 keV) with equipment of a secondary electron detector. Observations of Y_2_O_3_ in the as-deposited films were operated by a transmission electron microscopy (TEM, Tecnai TF20 TMP, Gatan Company USA) with an accelerating voltage of 200 kV. Cross-sectional TEM samples were prepared by a method combining the methods of mechanical thinning with Ar ion milling. 

Microhardness tests were carried out on surface of the films by the nanoindentation (Nano-indenter G200, Agilent, USA) with a strain rate of 0.05 s^−1^. The average hardness values were calculated from 15 separate indents with the depth range of 200–500 nm.

## 3. Results and Discussion

In order to investigate the effects of Y_2_O_3_ addition on the microstructure and mechanical properties of FeCr films, Y_2_O_3_/FeCr composite targets with 0, 1, 2 and 4 pieces of Y_2_O_3_ slices were used during sputtering (Figure 1a), and the resultant films were designated as FeCr, Y_2_O_3_-FeCr, 2Y_2_O_3_-FeCr, 4Y_2_O_3_-FeCr, respectively, herein. The concentration of Y_2_O_3_ in the FeCr films was analyzed by the inductive coupled plasma emission spectrometer (ICP), as shown in Table 1. It can be seen that as the number of Y_2_O_3_ slices is 0, 1, 2 and 4, the Y_2_O_3_ content is 0, 0.12, 0.23 and 1.97 wt.%, respectively. The abnormal increase of Y contents from 0.23 to 1.97 as the Y_2_O_3_ slice number increase from 2 to 4 was difficult to understand in terms of the sputtering area of the Y_2_O_3_ slice, but was tentatively attributed to the change in glow discharge mode induced by the Y_2_O_3_. 

Figure 1b,c show the surface and cross-section morphology of pure FeCr films and 2Y_2_O_3_-FeCr films, respectively. The large columnar crystal consisting of small columnar nanocrystals was observed in both the pure and 2Y_2_O_3_-FeCr films. For the 2Y_2_O_3_-FeCr film (Figure 1c), the number of grains per unit area was denser compared with that of the pure FeCr film (Figure 1b). In addition, the film thickness of the FeCr, Y_2_O_3_-FeCr, 2Y_2_O_3_-FeCr and 4Y_2_O_3_-FeCr films were 2.5, 1.5, 1.2, 1.0 μm, respectively, showing a decreasing trend with the increasing number of Y_2_O_3_ slice as shown in Table 1. The reason is that under the same sputtering conditions, the sputtering yield of Y_2_O_3_ ceramic is lower than that of the FeCr metal in the composite target.

Figure 2a shows the XRD patterns of FeCr-based films with four kinds of Y_2_O_3_ contents. All films exhibit the typical cubic ferrite phase structure, but the Y_2_O_3_-related diffraction peaks were not observed owing to their extremely low concentrations below the detecting limit of the XRD instrument. Based on the Scherrer equation *D* = *kλ*/*B*cosθ, where *D* is the coherent domain size, *k* is constant, *B* is the full width at half maximum (FWHW) and θ is the diffraction angle, the coherent domain size was roughly evaluated as about 7~12 nm (shown in Table 1) in terms of the FWHM of the (110) main diffraction peaks. It is worth pointing out that the coherent domain size *D* has a gradually decreasing tendency with an increase in Y_2_O_3_ content, which implies the grain refinement effect of Y_2_O_3_. This demonstrates that magnetron sputtering is a powerful technique for the preparation of ODS-nanocrystalline metal thin films.

To confirm the formation of the Y_2_O_3_, the XPS (X-ray photoelectron spectroscopy) analysis was conducted. As shown in Figure 2b, a specific electron binding energy was found to coincide with the energy value of Y_3d5/2_ electrons in molecular Y_2_O_3_. This result indicates that the Y elements exist in the form of Y_2_O_3_ in fabricated films, which is in line with the expectation of realizing oxide dispersion strengthening in FeCr based films by radio frequency magnetron sputtering (RFMS) methods.

Figure 3 displays the elemental distribution of the Fe, Cr, Y and EDS (Energy Dispersive Spectroscopy) spectrum of the 4Y_2_O_3_-FeCr film. The EDS mapping shows that all elements (Fe, Cr and Y) on the surface of the film are evenly distributed, though the signal of Y is weak due to the low content of Y, which is close to background level. The concentration of Ar is negligible, although argon is used as the working gas. The EDS spectrum obtained from the cross-section of the 4Y_2_O_3_-FeCr film in the line scanning mode demonstrates the existence of 1.58 wt.% Y and 0.42 wt.% O, corresponding to 2.0 wt.% Y_2_O_3_ in the film. In order to clarify how the element Y existed, the microstructures of pure FeCr and FeCr-0.23 wt.%Y_2_O_3_ were characterized using TEM and high-resolution TEM (HRTEM), as shown in Figure 4a–c. A large amount of nano-scale Y_2_O_3_ particles of 5–10 nm in diameter are uniformly dispersed in the volume of the FeCr-based films (Figure 4b). Based on the diffraction fringes in the high-resolution TEM (HRTEM) image of FeCr-0.23 wt.%Y_2_O_3_ (Figure 4c), the crystalline interplanar spacings are determined as 0.146 and 0.238 nm, in accordance with the (320) and (210) crystal plane of the cubic structure Y_2_O_3_ with the space group Ia3 (PCPDF card: 00-025-1200). These uniformly distributed Y_2_O_3_ nanoparticles would significantly strengthen the FeCr films. 

A nanoindenter technique was used to explore the microhardness of these ODS-FeCr-based films. Figure 5a displays the hardness vs. displacement curves of the FeCr-based films with different Y_2_O_3_ contents. The indentation depth *h* was controlled in the range of 0 to 500 nm in order to exclude the influence of the Si substrate. The effective nanoindentation hardness of the ODS-FeCr-based films deposited under different Y_2_O_3_ concentrations was calculated by averaging the relatively stable platform of hardness–depth profiles located in 100–500 nm, as shown in Figure 5b. It can be seen that the film hardness distinctly increases with the increase of the Y_2_O_3_ concentration, and the hardness value increases from 10.4 GPa to 16.5 GPa, exhibiting the obvious nano-oxide dispersion strengthened effect. Song et al. [27] raised an Orowan model demonstrating that the moving dislocation lines can be markedly hindered by the oxide particle. In the present Y_2_O_3_-doped FeCr films, the uniformly distributed nano Y_2_O_3_ particles act as an obstacle for the dislocation motion during deformation, which is beneficial for improving the mechanical property. Moreover, grain refining is another way to increase the hardness of hard films. Wang et al. demonstrated that the hardness (H) of a metallic material increases with a decreasing grain size d, according to the Hall–Petch relation H = H_0_ + kd^−1/2^ (k is constant) [28,29]. Grain refinement increased the number of grain boundaries, which improved the obstacle to material deformation and dispersed the stress concentration. Based on the analysis above, the extremely high hardness of Y_2_O_3_-doped FeCr films is attributed to the synergetic strengthen effect of dispersion strengthening and grain refinement strengthening. 

## 4. Conclusions

Y_2_O_3-_dispersed FeCr films were grown through a RF magnetron sputtering technique and the effects of Y_2_O_3_ addition on their microstructures and hardness were thoroughly investigated. The main results can be concluded as follows: (1)The coherent domain size of ODS-FeCr-based film is about 7~12 nm, and it decreases with an increase in Y_2_O_3_ content.(2)The addition of Y_2_O_3_ can obviously enhance the hardness of FeCr-based films, resulting in an extremely high value of 16.5 GPa.(3)The strengthening mechanism was from the nano-Y_2_O_3_ particle dispersion strengthening as well as the grain refinement strengthening.

## Figures and Tables

**Figure 1 nanomaterials-11-01754-f001:**
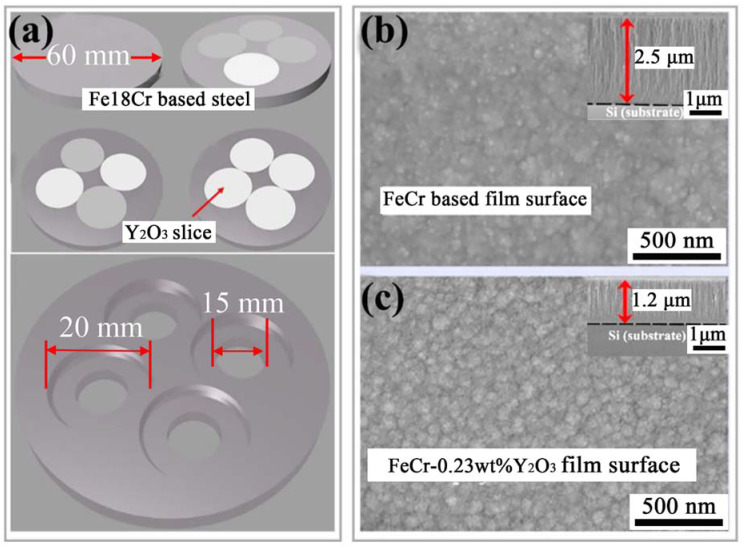
(**a**) Schematic of Y_2_O_3_/FeCr steel composited targets. White: Y_2_O_3_ slice; Dark: Fe/Cr metals slice; (**b**,**c**) surface and cross-section morphology of pure FeCr films with and without Y_2_O_3_ addition.

**Figure 2 nanomaterials-11-01754-f002:**
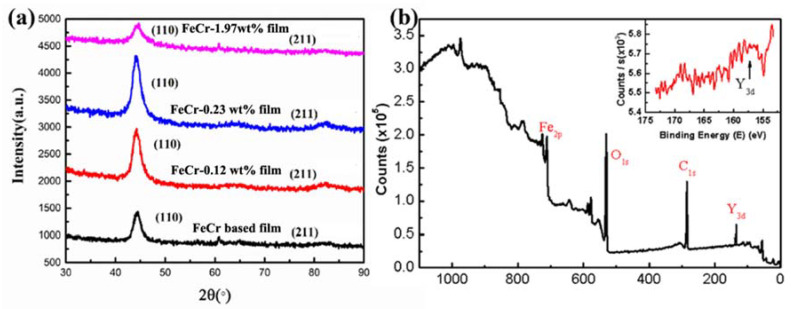
(**a**) X-ray diffraction patterns of FeCr films with different Y_2_O_3_ content; (**b**) XPS spectrum of 2Y_2_O_3_-FeCr (0.12 wt.% Y_2_O_3_) film.

**Figure 3 nanomaterials-11-01754-f003:**
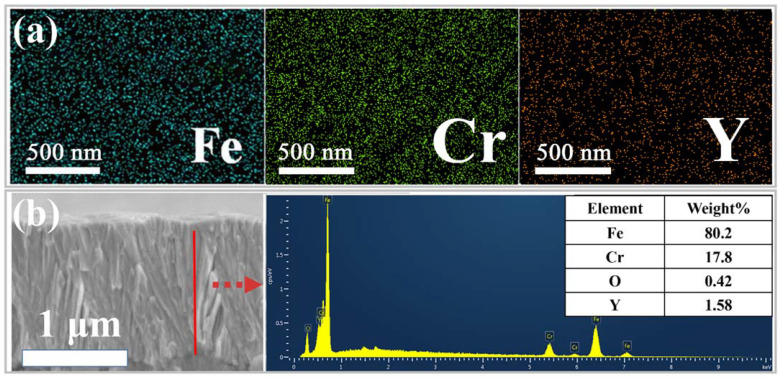
(**a**) 2D elemental mapping of Fe, Cr and Y in ODS-FeCr-based films prepared through sputtering FeCr target with 2 slices of Y_2_O_3_, (**b**) cross-section morphology and the corresponding EDS spectrum of FeCr-based films prepared through sputtering FeCr target with 4 slices of Y_2_O_3_.

**Figure 4 nanomaterials-11-01754-f004:**
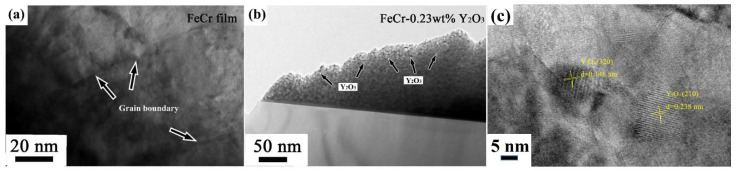
(**a**) Cross-sectional TEM images of FeCr films; (**b**) FeCr-0.23wt.% Y_2_O_3_; (**c**) HRTEM image of FeCr-0.23wt.%Y_2_O_3_.

**Figure 5 nanomaterials-11-01754-f005:**
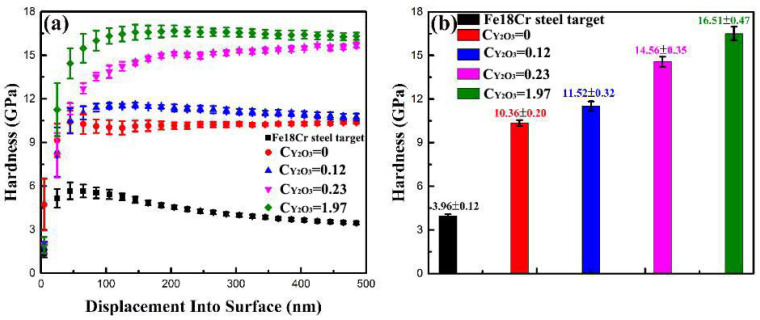
(**a**) Nanoindentation hardness (*H*)-depth (*h*) curves of the FeCr-based sheet and FeCr-based films with different Y_2_O_3_ content; (**b**) Hardness of the FeCr-based steel sheet and FeCr-based films with different Y_2_O_3_ concentration.

**Table 1 nanomaterials-11-01754-t001:** Y_2_O_3_ content, thickness and coherent domain size of Y_2_O_3_-doped FeCr films sputter deposited from FeCr target incorporated with 1, 2 and 4 pieces of Y_2_O_3_ slices, respectively.

Materials (Y_2_O_3_ Slices)	Y_2_O_3_ Content wt.%	Y_2_O_3_ Content at%	Thickness (μm)	Coherent Domain Size (nm)
0	0	0	2.5	12.65 ± 0.82
1	0.12	0.08	1.5	10.86 ± 0.94
2	0.23	0.14	1.2	9.63 ± 0.23
4	1.97	1.01	1.0	7.34 ± 0.25

## Data Availability

The data that support the findings of this study are available from the corresponding author, [author initials], upon reasonable request.

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
