# Peer review of "Effect of Nano-Y2O3 Content on Microstructure and Mechanical Properties of Fe18Cr Films Fabricated by RF Magnetron Sputtering"

_nanomaterials, 2021, doi:10.3390/nano11071754_

Round 1

Reviewer 1 Report

The manuscript describes the behavior of FeCr-based film as a function of Y2O3 content. As the Y2O3 increases, the grain size decreases, and the hardness increases. These materials are important to improve the nuclear reactor's performance since Y2O3 increases the irradiation resistance.  

The manuscript shows interesting results and good methodology. Nevertheless, some improvements can be done. Authors should emphasize the motivation of the work, which are the main achievements in comparison with other methodologies used to increase the irradiation resistance. Also, some comments about the improvement of the results, can be discussed, such as it is possible to increase the Y2O3 content?

Comments:

English language should be revised, for example:

Line 31: “higher service performance and reliability of for materials are …”

Line 112: “the surface was more smooth and…”

Figure caption, 2: “Atomic mapping of Fe…” suggestion: “2D elemental maps for Fe, Cr and Y.

Line 145: “film were shown” replace by “film are shown”, similar situations along with the document, for example, line 150

Line 147: “dispersed in the interior of FeCr based films” suggestion “dispersed in the volume of FeCr based films”

Other comments:

It is possible the incorporation of Ar atoms in the films? In many cases, the Ar (used during the growth process) can be incorporated in the film, affecting their properties.

Line 124 “All films exhibit the typical cubic ferrite phase structure and …” the typical ferrite phase of FeCr?

Which is the error when calculating the coherent domain size from XRD measurements?

Line 140, what does it mean :” The EDS mapping not only confirms the existence of Y elements”?

Figure 2. Include a color bar. It would be a good idea to include also an EDX spectrum. It seems the statistic for Y map is very low, very close to the background level. Which is the meaning of “FeCR+2Y2O3” in the Fe map?

Line 168 “define GBs”

Line 182. Films were not fabricated, films were grown, obtained…

Reviewer 2 Report

In this work, it is proposed to strengthen FeCr materials, which are used in various reactors, with the help of  Y2O3  additives. The material itself was obtained in the work using magnetron sputtering. It is shown that even a very small concentration of Y2O3  leads to hardening by a factor of 2.5, and when the maximum concentration of  Y2O3  is reached, the strength increases by more than 4 times.

In this work, it was found that the hardening mechanism is mainly associated with a decrease in the grain size from 12 to 7 nm.

The article is short and clear, so I believe it can be published without changes. 

Author Response

Comments:In this work, it is proposed to strengthen FeCr materials, which are used in various reactors, with the help of Y2O3 additives. The material itself was obtained in the work using magnetron sputtering. It is shown that even a very small concentration of Y2O3 leads to hardening by a factor of 2.5, and when the maximum concentration of Y2O3 is reached, the strength increases by more than 4 times.

In this work, it was found that the hardening mechanism is mainly associated with a decrease in the grain size from 12 to 7 nm.

The article is short and clear, so I believe it can be published without changes.

Responses: we deeply appreciate your encouragement and support.

Reviewer 3 Report

This manuscript reports the deposition of Y2O3-doped FeCr-based alloy films by an RF magnetron sputtering method, and their mechanical strength properties improved with the increase of the Y2O3 doping level. First of all, I could not understand the reason of employing thin films to study the Y2O3 doping effect to the alloy. The authors write that the grain size can be controlled in thin films (l.61). However, the fields of application are different between thin films and bulks (Obviously, thin films cannot be the material for construction of nuclear reactors). It would be understandable if one uses the thin film form as a platform of the basic research to investigate the mechanism of Y2O3-induced strenghtening, still there is no such discussion in the manuscript. It is difficult for me to recommend the manuscript for publication on journal Nanomaterials.

References on Y2O3-induced strenghtening on metals should be enriched. Besides Ref.16, I found some articles as below.
Y2O3-doped W-Ti alloys, 10.1016/j.jnucmat.2010.07.016
Y2O3-doped Mo, 10.1016/j.jmst.2021.01.064
Y2O3-doped Ti-Al-Cr-Nb alloy, 10.1016/j.msea.2021.140952

I believe that the authors’ method of controlling the film composition by changing the number of Y2O3 “slices” on the target holder is effective. Nevertheless, the relationship between the number of slices and the film composition is strange. Although the Y2O3 composition (in at%) seems proportional to the number of slices until two slices, it drastically increases (1.01%) at four slices. This is difficult to explain with the drop of exposing area of the FeCr holder. 

l.68 “Fe18Cr-based”: Describe the composition of the alloy as precise as possible.

l.70 “Si single crystal sheets” : “wafers” could be more popular than “sheets”. 

l.73 “rotating target”: It is quite exceptional, since the targets are generally water-cooled in sputtering deposition systems. I would like to read more explanation how the rotation and water-cooling of targets are possible simultaneously (or if the target is not water-cooled?). 

ll.91-97: The preparation process of TEM samples described here is popular and not necessary  in the article.

l.102 “ICP”: Spell it out at the first appearance.

Ll.112-113 “For … pure FeCr film.”: It is difficult to learn from Figs. 1b and 1c that the doped film has smoother surface or denser structure than the pure film (For me, it seems to be rather opposite). The authors should show evidences, such as AFM images, that supports their claim.

Fig. 1c shows a label “FeCr-0.12wt% Y2O3…”: According to the main text, this film was prepared with two Y2O3 slices. If so, the Y2O3 content of this film should be 0.23 wt% (Table 1), which contradicts with the label.

Fig. 1e: The texts are too small and the resolution is too low, resulting in some texts impossible to read. At first glance it was not clear whether the vertical bar under the label “Y3d” label in the main XPS spectrum was a peak or an arrow. I would like to suggest the authors to make Fig.1(b+c), Fig. 1d, and Fig. 1e independent figures (namely, Fig. 2(a+b), Fig. 3, and Fig. 4) with larger sizes.

Fig. 2: Signals in the Y mapping image is quite weak, not distinguishable from noise, and cannot be the evidence of existence of Y. 

Fig. 3b and 3c: I cannot see which structure the authors regard as Y2O3 particles.

The writing contains a number of grammatical errors and unnatural expressions. The manuscript should be revised with a help of a native speaker of English. Let me point out the errors only on page 1.
Title & l.14 microstructure, mechanical —> microstructure and mechanical
l.13 different Y2O3 content FeCr-based films —> FeCr-based films with different Y2O3 content
l.15 average —> averaged
l.25 twenty century —> twentieth century

Round 2

Reviewer 1 Report

The manuscript has been improved, but some questions are still open.

A second chance to review the manuscript is good since new comments and suggestions can arise.

The manuscript still contains a number of grammatical errors, it should be revised with a help of a native speaker of English.

Check the format, for example, the size of the letters.

Check the position of the figure and table captions.           

  • Please, remove “etc” from the abstract.
  • About the presence of Ar in the films. If the authors have checked that the concentration of Ar is negligible, it should be included in the manuscript.
  • About the coherent domain size from XRD. I cannot find the error added in the Table.
  • About EDS and 2D mapping from EDS. EDS can confirm the existence of Y atoms. I insist that include an EDS spectrum can be useful to check the signal of the elements (confirming also that Ar is not present). Also, a color bar should be included in figure 2. It seems that maps from Fe and Cr do not have enough statistics. Remove “FeCr+2Y2O3” from the Fe map. It seems that Fe maps correspond to this compound which is not the case.

Furthermore, which results can be extracted from EDS maps? The homogeneous distribution of Fe and Cr? The presence of Y was already confirmed ICP, XPS (from author response) and TEM…

  • About GBs, for “ grain boundaries”. What is the purpose of defining a new abbreviation (not universal) if it is only used once in the text?

Author Response

Many thanks for your suggestion, we have carefully revised our manuscript according to your comments, and the detailed response are listed below. We are expecting to receive good news and of course are total at your disposal.

Comment 1: The manuscript still contains a number of grammatical errors, it should be revised with a help of a native speaker of English.

Responses: the whole manuscript has been carefully gone through and revised accordingly.

Comment 2: Check the format, for example, the size of the letters.

Responses: the format has been carefully checked, with the unified font and size of letters, and interspace of paragraph.

Comment 3: Check the position of the figure and table captions.

Responses: the position of figure and table captions have been checked and revised in the revised manuscript as highlighted in yellow.            

Comment 4: Please, remove “etc” from the abstract.

Responses: “etc”in the abstract has been removed in the revised manuscript.

Comment 5: About the presence of Ar in the films. If the authors have checked that the concentration of Ar is negligible, it should be included in the manuscript.

Responses: “the concentration of Ar is negligible” has been incorporated into the revised manuscript and highlighted in yellow.

Comment 6: About the coherent domain size from XRD. I cannot find the error added in the Table.

Responses: I forget to add the revised Table 1 with error into the manuscript last time, but I have incorporated it into the revised manuscript and highlighted in yellow this time.

Comment 7: About EDS and 2D mapping from EDS. EDS can confirm the existence of Y atoms. I insist that include an EDS spectrum can be useful to check the signal of the elements (confirming also that Ar is not present). Also, a color bar should be included in figure 2. It seems that maps from Fe and Cr do not have enough statistics. Remove “FeCr+2Y2O3” from the Fe map. It seems that Fe maps correspond to this compound which is not the case.

Responses: one EDS spectrum as Fig.3b has been added into the revised manuscript, which clearly demonstrates the existence of Y and the absence of Ar.

Comment 8: Furthermore, which results can be extracted from EDS maps? The homogeneous distribution of Fe and Cr? The presence of Y was already confirmed ICP, XPS (from author response) and TEM.

Responses: the EDS mapping is adopted to illustrate how the elements distributed in the films. Hopefully, the uniform distribution of Fe, Cr, and the local enrichment of Y and O. But due to the resolution of EDS, only the first part is observed.

Comment 9: About GBs, for “ grain boundaries”. What is the purpose of defining a new abbreviation (not universal) if it is only used once in the text?

Response: the abbreviation of GBs has been deleted in the revised manuscript. 

Reviewer 3 Report

The second version was improved through revision after the comments of the reviewers to some extent. At this point I came to think that this manuscript has a chance of being accepted for publication. However, there are still some unsolved problems and some points to be reconsidered.

ll.72-78: The sentence is too long. The sentence pattern is incoherent. I would suggest the authors to omit ll.72-73 and to break the sentence at the end of l.76.

l.80: I now understand that Fe18Cr is a term popularly used in the field of ​​alloys. If the authors consider that all the readers would understand it as a Fe-Cr alloy with 18 at% Cr, no need in adding “(at%)”.

l.109: All common nouns should be written in lowercase.

l.119 “the surface was much denser”: This expression is not clear. "the number of grains per unit area was denser” would be would be more understandable.

Fig. 2a: If the four profiles are distinguished by the Y2O3 contents, the composition should be described in the legend of the figure, as was done in the previous version. Or, if they are distinguished by the number of Y2O3 slices as in the legend, the caption should describe as “prepared with different Y2O3 slices” and, in addition, the legends should be written not in simple numbers but as “4 slices”.

l.151 “… film sputtered by … target” is strange. It should be written as, for example, “… film prepared with… target” or “… film prepared by sputtering … target”.

Figs. 4(b) and 4(c): I still do not understand which parts of (b) and (c) correspond to Y2O3 grains. Do “a lot of white tiny particles” mean the points attached to the tips of the black arrows? How do authors know that they are Y2O3? According to Fig. 4(d), Y2O3 particles have a size of about 10 nm. If so, they should appear in (b) as a region with a certain size rather than as a point.
I would suggest the authors to show this figure to their colleagues around and ask them if they can figure out which part is Y2O3. I would withdraw my comment if everyone can correctly understand what the authors mean. If this is not the case, the authors should seek a better way of expression that conveys their intention more clearly. I do not even think it is necessary to show TEM images except (d).

Author Response

Dear reviewer

Many thanks for your suggestion, we have carefully revised our manuscript according to your comments, and the detailed response are listed below. We are expecting to receive good news and of course are total at your disposal.

Comment 1: line 72-78: The sentence is too long. The sentence pattern is incoherent. I would suggest the authors to omit ll.72-73 and to break the sentence at the end of l.76.

Responses: the sentence 72-78 has been revised as “It is known that thin films and bulks materials have different application field. The thin films can`t be used as the construction materials of nuclear reactors. But the FeCr based thin film can be adopted as a platform to model the nanocrystalline metal materials with an extremely high volume fraction of grain boundaries and to investigate the mechanism of Y2O3-induced strengthening. Therefore, in this work, the magnetron sputtering ODS-FeCr films were prepared and researched in order to provide a beneficial guidance for the research and development of fission reactor structural materials in the future”, and incorporated into the revised manuscript and highlighted in yellow.

Comment 2: line80 I now understand that Fe18Cr is a term popularly used in the field of ​​alloys. If the authors consider that all the readers would understand it as a Fe-Cr alloy with 18 at% Cr, no need in adding “(at%)”.

Responses: thanks for your understanding and support.

Comment 3: line 109 All common nouns should be written in lowercase.

Responses: “the Inductive Coupled Plasma Emission Spectrometer (ICP)” has been changed into “the inductive coupled plasma emission spectrometer (ICP)” in the revised manuscript and highlighted in yellow.

Comment 4. Line 119 “the surface was much denser”: This expression is not clear. "the number of grains per unit area was denser” would be would be more understandable.

Responses: many thanks for your excellent suggestion. We have used “the number of grains per unit area was denser” in the revised manuscript and highlighted in yellow.

Comment 5: Fig. 2a: If the four profiles are distinguished by the Y2O3 contents, the composition should be described in the legend of the figure, as was done in the previous version. Or, if they are distinguished by the number of Y2O3 slices as in the legend, the caption should describe as “prepared with different Y2O3 slices” and, in addition, the legends should be written not in simple numbers but as “4 slices”.

Responses: in Fig.2a the four profiles are distinguished by Y2O3 contents, and therefore the composition has been described in the legend of the figure as was done in the first version of the manuscript.

Comment 6: line 151 “… film sputtered by … target” is strange. It should be written as, for example, “… film prepared with… target” or “… film prepared by sputtering … target”.

Responses: the sentence has been changed into “film prepared through sputtering the composite target of FeCr with 2 slices of Y2O3” in the revised manuscript and highlighted in yellow.

Comment 7: Figs. 4(b) and 4(c): I still do not understand which parts of (b) and (c) correspond to Y2O3 grains. Do “a lot of white tiny particles” mean the points attached to the tips of the black arrows? How do authors know that they are Y2O3? According to Fig. 4(d), Y2O3 particles have a size of about 10 nm. If so, they should appear in (b) as a region with a certain size rather than as a point.
I would suggest the authors to show this figure to their colleagues around and ask them if they can figure out which part is Y2O3. I would withdraw my comment if everyone can correctly understand what the authors mean. If this is not the case, the authors should seek a better way of expression that conveys their intention more clearly. I do not even think it is necessary to show TEM images except (d).

Responses: there is an error in the scale bar in the previous manuscript, we have revised it correctly. In addition, the Fig.4 has been re-arranged and re-organized our language in order to express our thought more clearly. But I am still very sorry for the relatively poor quality of TEM image partially due to the limit of our TEM equipment. We will specially investigate the distribution of Y2O3 in the FeCr films in our next work.